# Puppies Raised during the COVID-19 Lockdown Showed Fearful and Aggressive Behaviors in Adulthood: An Italian Survey

**DOI:** 10.3390/vetsci10030198

**Published:** 2023-03-05

**Authors:** Luigi Sacchettino, Claudia Gatta, Andrea Chirico, Luigi Avallone, Francesco Napolitano, Danila d’Angelo

**Affiliations:** 1Department of Veterinary Medicine and Animal Production, University of Naples Federico II, 80137 Naples, Italy; 2Department of Psychology of Development and Socialization Processes, “Sapienza” University of Rome, 00185 Rome, Italy; 3CEINGE-Biotecnologie Avanzate Franco Salvatore, 80145 Naples, Italy

**Keywords:** dog personality questionnaire, dog behavior, human–dog relationship, socialization

## Abstract

**Simple Summary:**

The COVID-19 lockdown had a greater impact on growing individuals who were not fully accustomed to the relational experiences required for an appropriate socialization process. The purpose of our retrospective research was to investigate, through an online survey, the negative impact of restrictions experienced by puppies upon the adult dogs’ personalities. The present study highlights the importance of giving full attention to the relational needs of dogs during their early growth phases. In order to reduce the risk of episodes of aggression and fearfulness as well as to improve the wellbeing of animals raised under lockdown restrictions, we posit that a close monitoring approach should be adopted by veterinary behaviorists who can provide personalized rehabilitating therapies; this may be beneficial for both pets and humans.

**Abstract:**

During the COVID-19 pandemic, the Prime Minister’s decree issued by the Italian government on 9 March 2020, known as “#Iorestoacasa” (I stay at home), required everyone to stay indoors, with a few exceptions, from 11 March to 3 May 2020. This decree had a substantial impact on the mental health of both people and dogs. We carried out a national survey to compare the personalities of adult dogs who were puppies in lockdown (from March to May 2020) with those of adult dogs born after that time (from June 2020 to February 2021). Our results highlighted a significant increase in personality traits related to fear and aggression in dogs who experienced lockdown restrictions during their socialization period, further confirming that the COVID-19 pandemic strongly impacted the behavioral development of dogs. Therefore, it may be advantageous for these dogs to be closely monitored by veterinary behaviorists and receive specialized behavioral rehabilitation therapy to lower the risk of episodes of aggression and fearfulness as well as to increase the wellbeing of dogs raised under social restrictions.

## 1. Introduction

During the COVID-19 pandemic, the Italian government issued a Prime Minister’s decree on 9 March 2020; the decree, known as “#Iorestoacasa” (I stay at home), required everyone to stay indoors, with a few exceptions, from 11 March to 3 May 2020 [1]. Such essential measures impacted people’s physical and emotional health worldwide; alongside the he0alth issues caused by SARS-CoV2, the COVID-19 pandemic affected several key aspects of everyday life, due to the closure of many factories, schools, academic institutions, and other learning spaces [2,3,4,5,6]. However, the presence of companion animals (especially dogs) in the home turned out to play a key role in human wellbeing, as pet owners experienced evident improvements in their mental health and had reduced perceptions of loneliness [7,8,9]. In this respect, Olivia and Johnston proposed that pets provide opportunities for their owners to relax and calmly carry out their normal routines [10]. On the other hand, lockdown restrictions suddenly stopped or changed animals’ normal activities; they were only allowed to take short walks near their homes and could not socialize with other dogs or people [11,12,13,14,15,16], thus making them more stressed and prone to be incorrectly managed [17].

This is an issue that should be considered as part of the human–animal relationship, since companion animals are generally shaped by their owners’ habits. The COVID-19 pandemic certainly worsened some interpersonal aspects of this relationship, creating increased levels of frustration (due to the fact that pet owners generally worked from home), disrupting daily physical activities, exacerbating pre-existing behavioral issues, and limiting access to animal-related services outside the home (veterinary care, behavioral counseling, training courses) [18]. As suggested in our previous study, dogs experiencing lockdown restrictions appeared to be at even greater risk of behavioral disorders associated with social low stimulation and/or the constant presence of family members in the house, which increased separation-related problems [19]. In this regard, several studies have shown that the development of well-adjusted adult dogs, who exhibit few undesired behaviors, is greatly influenced by puppy socialization practices, which eventually result in a positive, long-lasting bond with their owners [20,21,22,23,24]. In many mammalian species, the first year of life is divided into three almost distinct developmental phases, which can be used to explain individual variability in behavior and performance: the primary, socialization, and juvenile periods. The “primary” or “neonatal” period, lasting from 3 to 16 days of life, relies on an emotional connection between mother and puppies. It is necessary that the female takes care of the cubs by licking, stroking, and squeezing them and responding to their calls [25]. The second phase, known as the “socialization” period, lasts from the time a pup is about 3 weeks old, until they are aged from 3 to 14 weeks old. Finally, the “juvenile” period, spanning from the end of socialization to sexual maturity (one year, on average) is characterized by the functional development of basic learning abilities and skills in environmental exploration, social contact, and avoidance [26]. During the socialization process, companion dogs should be exposed early to those situations, including animals, things, sounds, and textures, as well as people of different ages, genders, and races, that will be commonly encountered in their lifetimes [25]. Therefore, within a few days from birth, breeders should start a procedure that is tailored to the animals’ specific ages; this process needs to be continued by subsequent owners up to adulthood, to provide animals with a variety of experiences throughout their lifetimes [20]. The importance of exposing puppies to social and environmental cues during specific time windows has been widely documented, as they take more time to learn and to respond correctly to other animals or people during adulthood [21,25,27,28]. Brand et al. (2020) raised concerns about the socialization abilities of puppies aged from two to five months (which normally represents a critical period for emotional and behavioral development) during the pandemic lockdown, and suggested that limited exposure to novel experiences at that time may increase the likelihood of dogs exhibiting non-social and social fears (such as, respectively, phobias of loud noises and of the presence of other dogs or strangers) [29]. Thus, the COVID-19 pandemic restrictions gave rise to several long-term issues regarding the socialization process of puppies and its impact on their behaviors when they become adults. Therefore, this study aimed to better elucidate the potential impact of different environmental contexts during the socialization process upon dog personalities. The first of its kind in Italy, this study conducted a retrospective analysis of personalities of adult dogs who grew up during the COVID-19 pandemic lockdown, compared to animals born and raised after these restriction policies. 

## 2. Materials and Methods

Data were obtained via an online survey written in the Italian language and administered between February and September 2022 (see Appendix A for details). Participants were recruited using Facebook, Instagram, and WhatsApp, and were informed about the general purpose of the study and their rights to anonymity before joining the survey. The survey emerged from a multidisciplinary study resulting from a collaboration between the Department of Veterinary Medicine and Animal Production of the University of Naples Federico II and the Department of Psychology of Processes of Development and Socialization of the University of Rome La Sapienza. It consisted of 73 items and was divided into 2 parts, the first focusing on general and demographic information about dogs and their owners, the second a short-form Dog Personality Questionnaire [30]. The time needed to complete the survey was approximately 10 min. The collected data were coded and processed anonymously. Questionnaire replies were collected using the online survey platform Qualtrics^®^ (Qualtrics Software Company, Provo, UT, USA).

### 2.1. General and Demographic Information on Dog Owners and Their Dogs

These twenty-eight items were related to dog owners (i.e., “Age of owner in years”, “Sex at birth”, “Job”, “Level of education”, “Composition of the family unit”), household composition (i.e., “Size of dwelling”, “Presence of balcony/terrace/outdoor spaces”), dogs (i.e., “Sex of the dog”, “In what way did you adopt your dog?”, “How old was the dog at the time of adoption?”, “How many dogs live in the house?”), their management (i.e., “Were there any problems managing the dog after adoption?”, “Does the dog suffer from organ pathology?”, “Did the dog attended a dog trainer center?”). This part was helpful in defining the two study groups considered in the research (see Appendix A for further details). 

### 2.2. Measures

To test our hypothesis, we administered an adapted, well-validated version of the short-form Dog Personality Questionnaire [30]. The questionnaire consists of 45 items, divided into 5 factors and several facets: fearfulness (i.e., “Dog is shy”, “Dog behaves fearfully towards unfamiliar people”, “Dog exhibits fearful behaviors when restrained”, “Dog behaves fearfully when groomed”); aggression towards people (i.e., “Dog behaves aggressively towards unfamiliar people”, “Dog shows aggression when nervous or fearful”, “Dog behaves aggressively during visits to the veterinarian”, “Dog behaves aggressively in response to perceived threats from people”); aggression towards animals (i.e., “Dog is friendly towards other dogs”, “Dog behaves aggressively toward dogs”, “Dog likes to chase squirrels, birds, or other small animals”,” Dog likes to chase bicycles, joggers, and skateboarders”); activity/excitability (i.e., “Dog is boisterous”, “Dog is curious”, “Dog seeks constant activity”), and responsiveness to training (i.e., “When off leash, dog comes immediately when called”, “Dog is able to focus on a task in a distracting situation”, “Dog leaves food or objects alone when told to do so”), as shown in Table 1. Dog owners were invited to answer with one of the seven statements: disagree strongly, disagree moderately, disagree slightly, neither agree nor disagree, agree slightly, agree moderately, agree strongly. Two English–Italian bilinguals translated the survey into Italian from English using standardized back-translation procedures [31]. For our purposes, we did not use the items associated with the “trainability” factor. The items were selected considering both the original factor loadings and the context of quarantine, where it was forbidden to go outside for most activities (e.g., going to dogs’ training centers).

### 2.3. Statistical Analysis

We performed all of the statistical analyses presented above using the R language v.3.6.3 and the RStudio environment v.1.2.5033 [28], employing a statistical significance at α = 0.05. The main current study library was ‘psych’ [29,30]. Firstly, we computed descriptive statistics to present sample characteristics. Using the ‘MVN’ library [31], we checked the items’ distribution to verify the normality of the data. The internal consistency approach (Cronbach’s alpha and McDonald’s Omega) were used for reliability estimations. According to the scientific literature, a value over 0.70 can be considered satisfactory [32].

#### 2.3.1. Construct Validity

An explorative factorial analysis was conducted with oblique rotation (oblimin) on the aggregated item composing each facet, excluding the facet “responsiveness to training”, which lay outside the scope of the present study. The factor number was identified based on Horn’s Parallel Analysis [33]. In Horn’s Parallel Analysis [34], the observed eigenvalues extracted from the correlation matrix are evaluated with those obtained from simulated uncorrelated normal variables. This method uses a Monte-Carlo simulation process, considering “expected” eigenvalues derived by simulating normal random samples that coincide with the actual data in terms of sample size and the number of variables [35]. A factor-loading coefficient of 0.30 or higher was considered adequate [36]. The fit indices results were evaluated following the conventional criteria [37]: an RMSEA value below 0.06 and an SRMR value below 0.08. Based on how many components make up the scale, we estimated that the sample size required for factorial analysis would be a minimum of 100 patients [38].

#### 2.3.2. Differences between the Means

The dependent measures, namely, the factors revealed by the construct validity analysis, were analyzed by means of a paired-sample t test, considering the two study groups: “puppy lockdown”, or “dogs who were puppies during the quarantine”, and “puppy post-lockdown”, or “dogs born and matured after quarantine”. 

Data were analyzed using Jamovi version 2. The level of significance was set at α = 0.05; meanwhile, α levels between 0.05 and 0.10 were considered to be marginally significant. 

## 3. Results

The total number of people who visited the survey website was 405; among them, 310 fully completed the questionnaire, and therefore, represented the sample group for our analysis. Missing values from 49 people who partially responded to the test were processed listwise. The sample size consisted of 80.08% women and 19.921% men. The participants were 30–40 years old (30.15%), 20–30 years old (29.01%), 40–50 years old (25.19%), 50–60 years old (11.07%), 60–70 years old (4.20%), and over 70 years old (0.38%). The educational level of the sample group was 54.02% college, 42.91% high school, and 3.07% primary school. We found that the sample group was made up of retired people (3.09%), self-employed people (8.88%), people who work with dogs (12.36%), students (16.22%), freelance workers (16.60%), employed people (31.66%), and others (11.20%). Most of the respondents had balconies or terraces associated with their homes (90.31%), while others had shared or private green areas available (71.81%). The dogs included in the study were from shelters (19.20%), adopted from previous owners (20.80%), found as strays (11.20%), purchased from private people (15.20%) or dog farms (25.20%), bought in stores (2%), or acquired by other means (6.40%). The age of the animals when they were adopted was one month (9.39%), two months (47.35%), three months (31.02%), four months (3.27%), or five months (4.90%), while 4.08% of them were born in their current homes. No problems in dog management were experienced by the owners after adoption in 81.78% of cases. Most of the owners (91.98%) stated that their dogs did not show any relevant clinical variations. Data analyzed for the demographic features of both dogs and their owners did not reveal any significant effects related to the dogs’ gender and age of adoption, or to the presence of any other clinical variations, the number of people they live with, the number of dogs in the family, or the home size, gender, and education level of the owners (Table 2).

### 3.1. Construct Validity

Before conducting EFA, we performed both Bartlett’s test of sphericity and the Kaiser–Meyer–Olkin measure of sampling adequacy (KMO). The correlation matrix showed that most items resulted in a relationship greater than 0.35. Bartlett’s test of sphericity was significant (*p* < 0.001) and the KMO value was >0.6 (KMO = 0.70), indicating that factor analysis was appropriate for the data [39]. The parallel analysis (5000 parallel datasets, α = 0.01) revealed a three-factor solution. The three factors accounted for 56.0% of the total variance. As shown in Table 1, the first and the third factors that emerged were coherent with the original asset of the scale (11), while the facets of “Aggression towards people” and “Aggression toward dogs” loaded on the same factor. Fit indices for the EFA were considered acceptable (RMSEA: 0.05, TLI: 0.95, SRMR: 0.03), which indicated a consistent factor structure. Some of the facets were discarded (i.e., “fear of dogs”, “aggression against dogs”) from the analysis because they did not reach the minimum of 0.30 factor loading (Table 3).

### 3.2. Perceived Difference between “Puppy Lockdown” and “Puppy Post-Lockdown”

The normality of the data was confirmed by Skewness and Kurtosis values within the range of −1 and +1. The results of the one-paired sample *t*-test on the factors that emerged from the construct validity analysis showed a significant difference between dogs raised during and after the quarantine period for the “Fear” factor; furthermore, a statistical trend (*p* = 0.057) was found for the differences in terms of aggressivity (see Figure 1 for descriptive statistics and graphical information). In order to descriptively understand the factors related to the emerging differences, a series of independent *t*-tests was also performed for the facets composing each factor (see Table 4 for a complete list of comparisons on the facets composing the factors).

## 4. Discussion

Dog personalities are a topic of public interest because they have a variety of real-world implications associated with the deep impact of both inter- and intra-specific relationships. The present study aimed to assess the detrimental effects of the pandemic lockdown upon adult dogs who were raised during a period of social and environmental restrictions. As expected, the most salient result is the significant increase in aggression and fear traits in these animals, when compared to age- and sex-matched animals who were born after the COVID-19 lockdown.

Our data showed a gender effect in the people who took part in the survey, since women were more prevalent than men, confirming previous findings [19,32,33]; this effect may be due to women’s greater inclinations to participate in online surveys in general, and to look after pets [34,35]. Our sample fell into the medium-high category in terms of educational attainment, with 54% of respondents having a college-level education. Accordingly, previous data pointed to a link between a person’s socio-cultural level and stronger resiliency towards difficulties because of their possession of more advanced cognitive tools [36], which could have a positive impact on their relationship with dogs and their ability to manage a dog’s behavioral problems. It is interesting that 12.36% of the respondents worked in the dog and veterinary fields, and are therefore taught to be more conscious of the relationship between humans and dogs [19]. Although there is a strong bond between humans and dogs, a number of dysfunctional issues can arise in animals, including excessive interspecies aggression, fear and anxiety, abnormal repetitive behaviors, and animal hoarding, which can lead to animals eventually being relinquished to shelters [37,38,39,40,41,42,43]. The pandemic lockdown has exacerbated these issues in the normal routines of both dogs and caregivers [8,12,15,44], who were forced to change their lifestyles, reducing the frequency and duration of walks and causing a pronounced lack of inter and intra-species socialization [19]. The data that we collected highlight the significant impact of lockdown restrictions in the “puppy lockdown” group, showing that these restrictions were associated with aggressive and fear-related personality traits; these results are in line with our previous finding that a lack of exposure to urban surroundings and non-domestic maternal environments in dogs between three and six months of life was related to aggressive behavior toward strangers and to avoidance behavior [45,46]. In keeping with recent reports described by Tulloch et al. (2021), our study showed a trend toward increased handling fear in lockdown puppies, a behavioral alteration quite similar to that sometimes observed in dogs during veterinary examinations [47,48,49,50,51]. This is not surprising, since veterinarian behaviorists advise that puppies should be aided in adjusting to contact with all areas of the body, comparable to what might be experienced with a child, by creating positive associations (e.g., with rewards or during play) while caressing the tail, ears, and body and holding the collar. Any type of physical punishment, threats with a hand, or forceful interactions (e.g., pinning, rolling over) should be avoided. Indeed, dogs must learn that the human hand is friendly and not to be feared (i.e., they are associated with treats, toys, and affection), and this process should be extended to other non-cohabiting humans, as well as to veterinarians [26]. 

These findings confirm the study of Brand et al., who reported that pandemic puppies in the UK were significantly less exposed to visitors in their homes and were also significantly less likely to have had a veterinary health check. It is likely that this lack of exposure to handling also occurred in Italy, given that veterinary assistance during the block period was only allowed for emergencies, as shown in our previous study [19]. Additionally, Boardman and Farnworth (2022) showed that most dogs experienced increased excitement towards people by showing fear-related behavior towards both people and dogs, especially once lockdown limitations were loosened [14]. In keeping with these findings, we found a trend towards fearfulness of people in our enrolled dogs, which constitutes a further altered personality trait. Fear in dogs is an undesirable trait that can bring about abandonment, aggression and, in the worst-case scenario, pose threats to public health worldwide, thus necessitating euthanasia [52,53]. This is confirmed by Palestrini et al. (2005), who showed that behavioral and physiological stress responses may also occur when dogs are handled by unfamiliar humans [54]. Lockdown puppies mostly grew up with few interspecies reference patterns, aside from their own caregivers, since social interactions were drastically disrupted; as such, they are not able to properly take advantage of socialization experiences from “unfamiliar” non-cohabiting humans. Moreover, some of the analyzed dogs displayed conspecific aggressions, behavior that was already documented in a previous report, where 14% of seized dogs had seriously wounded or killed another dog, most likely either because of poor interactions with other dogs during the socialization period [55], or because of the characteristics of intraspecific socialization and the age of adoption of the dogs involved. The primary socialization period for dogs begins at three weeks of age, peaks at six to eight weeks, and progressively reduces by twelve weeks [26,56,57]. In addition, puppies generally exhibit pro-social behaviors in both intra- and interspecific interactions during this period, as well as a decrease in their tendency to fear or avoid unfamiliar environments. Puppies’ propensity to approach strangers increases between three and five weeks of age but then decreases [28]. Our results showed that 47% of the puppies were adopted at the age of two months, confirming that their intraspecific socialization only partially took place before the adoption and the restriction period, according to Italian law, regarding conditional and urgent ordinances concerning measures for the identification and registration of the dog population (https://www.gazzettaufficiale.it/eli/gu/-2008/08/20/194/sg/pdf, accessed on 27 February 2023). However, intraspecies socialization requires frequent and repeated contact over time, at least until social maturation is achieved, as a lack of socialization during this sensitive period and during the rest of the dog’s life plays an important role in the development of behavioral problems in adulthood [20]. In support of this theory, recent research revealed that “pandemic puppies” were less likely to have gone to puppy training classes or have been exposed to strangers before the age of 16 weeks; this issue also occurred in Italy because lockdown restrictions affected dog centers, forcing them to shut down [29]. Lockdown puppies also exhibited increased non-social fearfulness attitudes; this is unsurprising, considering that a recent study reported that lockdowns in Spain led to an increase in the fear of loud or sudden noises [32]. Dogs with recurrent non-social fears generally experienced a poor socialization process during puppyhood, thus suggesting that puppies should be introduced to stimuli and conditions they are likely to encounter as adults, improving the plasticity of their behavior, in order to create well-balanced adult dogs who are able to cope with new contexts [56,57,58,59,60,61]. In the light of the severe developmental alterations in puppies born during COVID-19 pandemic lockdown, which have compromised their physiological behaviors, it is imperative that we find alternative strategies that allow “patients” to cope with such psychosocial disorders. In this respect, taking advantage of the expertise of both vet behaviorists and dog instructors, dog owners might be encouraged to perceive dogs’ emotional alterations as a vulnerability factor, in order to improve their consciousness of the dogs’ needs and to adopt a suitable management level. To accomplish this aim, well-trained owners may employ three complementary approaches, based on (1) the education level of the family, (2) modification of the environment and (3) modulation of the patient’s behavior. The ultimate success of treating these issues is directly related to the degree of the owner’s comprehension and compliance. Family members themselves should cooperate with the behavioral changes of both the patient and their cohabitants, allowing for a mutual improvement in social communication and signaling. Moreover, owners are encouraged to take note of what their own roles are, and to establish a fully empathic relationship with the dog. Environmental modification might be associated with the management of different parameters, and is aimed at reducing the performance or intensity of the pathologic behavior; techniques may involve (a) exposure to positive stimuli, (b) consistent and predictable consequences that use rewards, rather than punishment, to encourage desirable behaviors, offering adequate enrichment to meet the dogs, and (c) paying attention to the identification and removal (or reduction) of stressors that activate problematic traits [26,60]. Finally, based on our research experience at the academic teaching hospital, behavioral shaping might be improved through a tailored therapeutic intervention; such interventions are characterized by the use of olfactory and cooperative learning, as well as calming and relaxing activities, including resting on a mat, or chewing on an appropriate chew toy. It is worth noting that, with fearful and aggressive dogs, a therapeutic intervention constitutes a set of techniques and skills, as it should be shaped and tailored to each clinical case not only according to the diagnosis and evolution of the pathology, but also according to the animal’s resources and the family system.

## 5. Conclusions

Our findings confirm previous results about the importance of socialization for dog behavioral development, highlighting how confinement restrictions enforced due to the COVID-19 pandemic induced more aggressive and fearful personality traits in dogs. In order to reduce the risk of aggression and fearfulness and to increase the wellbeing of dogs raised under lockdown restrictions, we conclude that it may be beneficial to have them closely monitored by veterinary behaviorists and receive tailored rehabilitation therapy. Additionally, owners should be trained to respond to any future behavioral changes brought on by unexpected conditions which may disrupt the dynamic between dogs and their owners.

## Figures and Tables

**Figure 1 vetsci-10-00198-f001:**
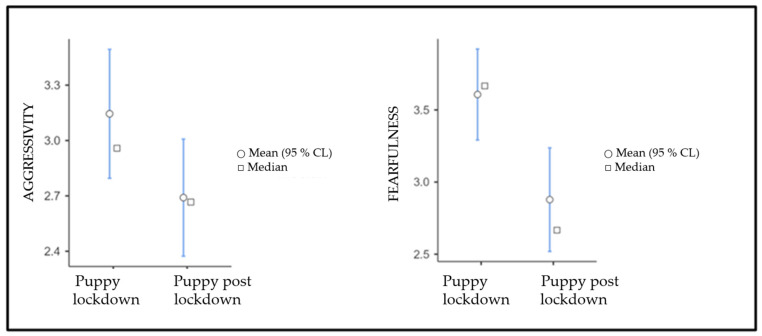
Differences in the aggressiveness and fearfulness traits in the “puppy lockdown” sample group.

**Table 1 vetsci-10-00198-t001:** Behavioral features considered for the enrolled dogs.

Factors (n)	Facet (n)
(1) Fearfulness	(1) Fear of People (2) Nonsocial Fear (3) Fear of Dogs (4) Fear of Handling
(2) Aggression against People	(1) General Aggression (2) Situational Aggression
(3) Activity/Excitability	(1) Excitability (2) Playfulness (3) Active Engagement (4) Companionability
(4) Responsiveness to Training	(1) Trainability (2) Controllability
(5) Aggression against Animals	(1) Aggression against Dogs (2) Prey Drive (3) Dominance over Other Dogs

**Table 2 vetsci-10-00198-t002:** Demographic table for the dogs and owners enrolled in the study.

	Puppy Lockdown (%)	Puppy Post Lockdown (%)
Enrolled dogs	55.81	44.19
Age at adoption (born at home)	3.79	3.81
Age at adoption (1 month)	9.09	10.48
Age at adoption (2 months)	45.45	50.48
Age at adoption (3 months)	33.33	27.62
Age at adoption (4 months)	3.03	3.81
Age at adoption (5 months)	5.30	3.81
Intact males	41.22	31.13
Neutered males	7.63	9.43
Intact females	28.24	27.36
Spayed females	22.90	32.08
Dogs suffering from organ pathologies	7.09	5.83
Dog owners’ sex (male)	23.78	16.22
Dog owners’ sex (female)	76.22	83.78
Dog owners’ education (primary school)	3.50	2.70
Dog owners’ education (high school)	47.55	36.04
Dog owners’ education (college)	48.95	61.26
Home size (≤65 s.m.)	10.64	13.64
Home size (from 66 to 110 s.m.)	54.61	36.36
Home size (from 111 to 150 s.m.)	25.53	31.82
Home size (>150 s.m.)	9.22	18.18
Number of people living with the dogs (from 2 to 6)	93.70	86.49
Additional dogs living with the enrolled dog (more than 1)	29.93	34.86

**Table 3 vetsci-10-00198-t003:** Factor loadings. Facet 1—excitability; Facet 2—playfulness; Facet 3—active engagement; Facet 4—companionability. The ‘Minimum residual’ extraction method was used in combination with an ‘oblimin’ rotation.

Factor Loadings				
	Factor			
	Fearfulness	Aggressiveness	Activity/	Uniqueness
			Excitement	
Fear of people	0.978			0.0825
Fear of handling	0.649			0.4853
Nonsocial fear	0.637			0.4764
Situational aggression		0.812		0.2461
Prey drive		0.711		0.4671
Dominance towards other dogs		0.692		0.5825
General aggression		0.416		0.2333
Playfulness			0.654	0.4789
Companionability			0.547	0.4656
Excitability			0.488	0.6065
Active engagement			0.349	0.7898

**Table 4 vetsci-10-00198-t004:** Independent samples *t*-test on the facets that emerged from the construct validity analysis. Significance < *p* a 0.05; >0.10 indicates a statistical trend close to significance.

Personality Traits	Statistic		df	*p*	Mean Diff.	SE Diff.	Effect Size (Cohen’s d)
Dominance towards other dogs	Student’s t	1.805	118	0.074	0.409	0.227	0.3299
Prey drive	Student’s t	0.863	190	0.389	0.155	0.180	0.1252
Aggression towards dogs	Student’s t	2.934	141	0.004	0.735	0.251	0.4927
General aggression	Student’s t	2.335	189	0.021	0.465	0.199	0.3393
Situational aggression	Student’s t	1.267	161	0.207	0.327	0.258	0.2006
Fear of Handling	Student’s t	2.691	186	0.008	0.588	0.219	0.3943
Fear of Dogs	Student’s t	1.523	186	0.129	0.298	0.195	0.2230
Nonsocial Fear	Student’s t	3.257	131	0.001	0.828	0.254	0.5685
Fear of People	Student’s t	1.802	178	0.073	0.421	0.234	0.2693

## Data Availability

Not applicable.

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
