# Peer review of "Puppies Raised during the COVID-19 Lockdown Showed Fearful and Aggressive Behaviors in Adulthood: An Italian Survey"

_vetsci, 2023, doi:10.3390/vetsci10030198_

Round 1
Reviewer 1 Report
The paper is well conceived, the introduction contains adequate information about the problem to be examined. The material and methods are described. The results are adequately tabulated, and the discussion follows the results obtained. Conclusions derived from the results obtained.
Comments:
Introduction
Line 70 – 72 the sentence must be reworded
Line 72-73 the sentence must be deleted or moved to the material and methods section
Line 73-78 this paragraph must be deleted, it is more appropriate for the conclusions section
Discussion
Line 218-220 this is not a part of the discussion section, this seems to be the aim of the study
Author Response
The paper is well conceived, the introduction contains adequate information about the problem to be examined. The material and methods are described. The results are adequately tabulated, and the discussion follows the results obtained. Conclusions derived from the results obtained.
Thank you very much for the kind comments; we edited the manuscript, making it easier to understand.
Comments:
Introduction
- Line 70 – 72 the sentence must be reworded
- Done it.
- Line 72-73 the sentence must be deleted or moved to the material and methods section
- Done it.
- Line 73-78 this paragraph must be deleted, it is more appropriate for the conclusions section
- We thank the reviewer for her/his suggestion. In the revised version of the manuscript, we deleted that paragraph.
Discussion
- Line 218-220 this is not a part of the discussion section, this seems to be the aim of the study
- Done it.
Reviewer 2 Report
It is not clear in the study whether there are differences between the groups (born during lockdown and those born after lockdown), whether the groups share the same characteristics of home size, number of people they live with, number of animals, gender and respondents' education ... these factors could interfere with the result. If the groups are not uniform in this sense, the authors cannot conclude whether the fact that led to the difference in behavior was being born/raised during the lockdown.
Author Response
Reviewer 2
- It is not clear in the study whether there are differences between the groups (born during lockdown and those born after lockdown), whether the groups share the same characteristics of home size, number of people they live with, number of animals, gender and respondents' education ... these factors could interfere with the result. If the groups are not uniform in this sense, the authors cannot conclude whether the fact that led to the difference in behavior was being born/raised during the lockdown.
- We’re grateful to the Reviewer for her/his thoughtful comments. Comparison between two groups (puppy lockdown vs puppy post lockdown) was not so evident in the description of survey, thus making difficult to underly the negative impact of social restrictions under COVID-19 pandemic. Therefore, in the revised version of the manuscript we implemented “Results” section with the demographic characteristics of the two samples. As she/he can see in the main text (Table 1), no main effect emerged from the comparison between home size, dogs’ gender, number of people they live with, number of dogs in family, gender’s owner, and respondents' education. We also improved the materials and methods, as suggested.
Reviewer 3 Report
This is a reasonable idea and analysis, although the findings are not particularly novel beyond context (ie that poor socialisation increases aggression and fear).
The main issue is the writing and data presentation, which are somewhat hard to follow, especially given substantial errors and excessively long sentences.
In the abstract it would be good to see some of the basic results that support your conclusions.
Please proofread.
"The Italian National report emphasized" (what is this report? Be specific.)
"paving the way to" (what are the novel perspectives?)
I think that educating owners may be a first step before behaviourists are involved (which would generally be in extreme cases).
L51: their animals for short walks
L53: social lives of both parties
L56: spelling error: outdoor
L56: "limited outdoor access and access to...etc" (delete "and to the outside" (L57))
L58: "in our" not "in a our"
L58: "[9], prior to 2 years of age, dogs appear"
L59: greater risk of what?
L60: "which increased separation related problems"
I am going to stop correcting spelling and syntax, but would encourage the authors to have the ms. reviewed in detail by a native English speaker. The phrasing, syntax and spelling require a lot of attention.
The introduction is very short and should be expanded to include more about what is currently known, both in terms of socialisation in general, it's impacts and the situation during lockdown.
L72-78: info about analysis, data collection, results and conclusions should not be placed in the intro. Delete and distribute in relevant sections.
L80: written in Italian
Briefly outline the sections of the survey suppl material is useful, but the main contexts should be in the body of the ms.
L86: "placed based composed" doesn't make sense
Place factors and facets into a table so they can be easily understood.
L100: why were these not used? They would seem relevant to the study.
Where were advertisements placed? Social media? Which platforms specifically?
L117-119: this sentence doesn't make sense.
L308: used 1dp for these (and all) values given the sample is 310 2dps are not required
Fig 1 is very small and can't be read. In B, Participants is spelled incorrectly.
Personally, I would encourage the use of tables rather than pie charts to represent the descriptive stats for owners and dogs.
L161: What is an organic disease (do you mean "organ"?)
L156: do you mean "found as strays (11.2%)"?
L178: whiihc were discarded? {Place them in the table and identify them using a symbol/footnote)
Fig 3: also small
L208: do you mean "with 54% having college-level education or higher"?
L211: "working dogs"
211: taught
:214-215: do you mean "relinquished to shelters"?
Delete phrase "where....exile"
L218: is this inter and intra-species socialisation?
L221: what are "restraint dogs"?
Author Response
Reviewer 3
Q: This is a reasonable idea and analysis, although the findings are not particularly novel beyond context (ie that poor socialisation increases aggression and fear).
A: We agree with the reviewer for her/his consideration. The present retrospective analysis stems from the observation of behavioral disorders, characterized by fear and reactivity in the dogs, who spent their childhood under pandemic restrictions, for which they came to our Teaching Hospital for behavioral examinations. Therefore, although we sought to confirm and highlight the potential correlation between current detrimental phenotypes observed dogs, and their previous “non-physiological” living conditions, we aimed at stressing the overall importance of providing puppies an integrated and balanced social development.
- The main issue is the writing and data presentation, which are somewhat hard to follow, especially given substantial errors and excessively long sentences.
- Thank the reviewer for her/his remark. In the new version of the manuscript we rephrased sentences according the suggestions. In addition, we requested editing from the Journal, by a native speaker to improve understanding.
- In the abstract it would be good to see some of the basic results that support your conclusions.
- We thank the reviewer. Here we rewrote the section, as suggested.
- "The Italian National report emphasized" (what is this report? Be specific.)
- We detailed what that report is.
- "paving the way to" (what are the novel perspectives?)
- We thank the Reviewer for this comment. We agree, in the bibliography there are several studies that show the importance of socialization upon the dog’s behavioral development. We started from the studies of Brand et al., that suggested majors shifts in the socialization and demographics of the puppies acquired during the 2020 phase of the COVID-19 pandemic in the UK, raising concerns for the future welfare of Pandemic Puppies. Our retrospective analysis aimed to investigate, through a personality test on adult dogs, how lockdown restrictions impacted the pandemic puppies.
- L51: their animals for short walks
- Done it.
- L53: social lives of both parties
- Done it.
- L56: spelling error: outdoor
- Done it.
- L56: "limited outdoor access and access to...etc" (delete "and to the outside" (L57))
- Done it
- L58: "in our" not "in a our"
- Done it.
- L58: "[9], prior to 2 years of age, dogs appear"
- Done it.
- L59: greater risk of what?
- We amended this sentence.
- L60: "which increased separation related problems"
- We amended this sentence.
- I am going to stop correcting spelling and syntax, but would encourage the authors to have the ms. reviewed in detail by a native English speaker. The phrasing, syntax and spelling require a lot of attention.
- We agree with the Reviewer. We revised the text and requested editing from a native speaker.
- The introduction is very short and should be expanded to include more about what is currently known, both in terms of socialisation in general, it's impacts and the situation during lockdown.
- We thank the reviewer for her/his kind comments. We followed the reviewer’s suggestion to further enhance the quality of the introduction.
- L72-78: info about analysis, data collection, results and conclusions should not be placed in the intro. Delete and distribute in relevant sections.
- We fixed it, as suggested.
- L80: written in Italian
- We fixed the mistake.
- Briefly outline the sections of the survey suppl material is useful, but the main contexts should be in the body of the ms.
- We fixed it, as suggested.
- L86: "placed based composed" doesn't make sense
- We amended the sentence.
- Place factors and facets into a table so they can be easily understood.
- We thank the reviewer. In the revised version of the manuscript, we added a new Table, which includes factors and parameters.
- L100: why were these not used? They would seem relevant to the study.
- We thank the reviewer for her/his question. For our purpose we did not use the items associated to the "trainability" factor. The items were selected considering both original factor loadings and context of quarantine, where it was forbidden to go out for any activities (e.g., going to dogs' training centers).
- Where were advertisements placed? Social media? Which platforms specifically?
- We thank the reviewer for her/his questions. In the revised version of the manuscript, we stated the required information.
- L117-119: this sentence doesn't make sense.
- Thanks’, we rewrote the sentence.
- L308: used 1dp for these (and all) values given the sample is 310 2dps are not required
- Done it.
- Fig 1 is very small and can't be read. In B, Participants is spelled incorrectly.
- We’re grateful to the reviewer. We amended the figure accordingly.
- Personally, I would encourage the use of tables rather than pie charts to represent the descriptive stats for owners and dogs.
- Done it.
- L161: What is an organic disease (do you mean "organ"?)
- We thank the reviewer. Organic disease means structural alterations in particular tissues/organs. However, to make the concept clearer, in the revised version of the manuscript we rephrased the sentence as “…did not show any relevant clinical alterations”.
- L156: do you mean "found as strays (11.2%)"?
- Yes, we do. We changed it, as suggested.
- Fig 3: also small
- We’re grateful to the reviewer. We amended the figure accordingly.
- L208: do you mean "with 54% having college-level education or higher"?
- We rewrote the sentence, accordingly.
- L211: "working dogs"
- We thank the Reviewer for her/his comment. Here we referred to the owners, who work in dogs’ care. We amended this expression in the new version of the manuscript.
Q.211: taught
- We fixed the mistake.
- 214-215: do you mean “relinquished to shelters”?
- Yes, we do. We changed it, as suggested.
- Delete phrase "where....exile"
- Done it.
- L218: is this inter-and intra-species socialisation?
- Yes, it is. We changed it, as suggested.
- L221: what are "restraint dogs"?
- We thank the question of the Reviewer. By “restraint dogs” we meant the group of dogs that experienced the restrictions of the Italian lockdown period (March-May 2020). We have better clarified this concept in the revised version of the manuscript.